# Epidemiological insights from a large-scale investigation of intestinal helminths in Medieval Europe

**Patrik G. Flammer**[1☯], **Hannah Ryan**[2☯], **Stephen G. Preston**[1], **Sylvia Warren**[1], **Renáta Přichystalová**[3], **Rainer Weiss**[4], **Valerie Palmowski**[5], **Sonja Boschert**[5], **Katarina Fellgiebel**[5], **Isabelle Jasch-Boley**[5], **Madita-Sophie Kairies**[5], **Ernst Rümmele**[4], **Dirk Rieger**[6], **Beate Schmid**[4], **Ben Reeves**[7], **Rebecca Nicholson**[8], **Louise Loe**[8], **Christopher Guy**[9], **Tony Waldron**[10], **Jiří Macháček**[3], **Joachim Wahl**[4,5], **Mark Pollard**[11], **Greger Larson**[2], **Adrian L. Smith**[1]*

1 Department of Zoology, University of Oxford, Oxford, United Kingdom, 2 Palaeogenomics & Bio-Archaeology Research Network, University of Oxford, Oxford, United Kingdom, 3 Department of Archaeology and Museology, Masaryk University, Brno, Czech Republic, 4 Landesamt für Denkmalpflege Baden-Würtemberg, Esslingen am Neckar, Germany, 5 Altertums—und Kunstwissenschaften, University of Tübingen, Tübingen, Germany, 6 Archäologie und Denkmalpflege der Hansestadt Lübeck, Lübeck, Germany, 7 York Archaeological Trust, York, United Kingdom, 8 Oxford Archaeology Ltd., Oxford, United Kingdom, 9 Worcester Cathedral, Worcester, United Kingdom, 10 University College London, London, United Kingdom, 11 Research Laboratory for Archaeology and the History of Art, University of Oxford, Oxford, United Kingdom

☯ These authors contributed equally to this work.
* adrian.smith@zoo.ox.ac.uk

**Editor:** jong-Yil Chai, Seoul National University College of Medicine, REPUBLIC OF KOREA

**Data Availability Statement:** Sequences have been deposited on the NCBI genbank under accession numbers: MT703650 - MT703665 and MH599138- MH599880.

## Abstract

Helminth infections are among the World Health Organization's top neglected diseases with significant impact in many Less Economically Developed Countries. Despite no longer being endemic in Europe, the widespread presence of helminth eggs in archaeological deposits indicates that helminths represented a considerable burden in past European populations. Prevalence of infection is a key epidemiological feature that would influence the elimination of endemic intestinal helminths, for example, low prevalence rates may have made it easier to eliminate these infections in Europe without the use of modern anthelminthic drugs. To determine historical prevalence rates we analysed 589 grave samples from 7 European sites dated between 680 and 1700 CE, identifying two soil transmitted nematodes (*Ascaris* spp. and *Trichuris trichiura*) at all locations, and two food derived cestodes (*Diphyllobothrium latum* and *Taenia* spp.) at 4 sites. The rates of nematode infection in the medieval populations (1.5 to 25.6% for *T. trichiura*; 9.3–42.9% for *Ascaris* spp.) were comparable to those reported within modern endemically infected populations. There was some evidence of higher levels of nematode infection in younger individuals but not at all sites. The genetic diversity of *T. trichiura* ITS-1 in single graves was variable but much lower than with communal medieval latrine deposits. The prevalence of food derived cestodes was much lower (1.0–9.9%) than the prevalence of nematodes. Interestingly, sites that contained *Taenia* spp. eggs also contained *D. latum* which may reflect local culinary practices. These data demonstrate the importance of helminth infections in Medieval Europe and provide a baseline for studies on the epidemiology of infection in historical and modern

**Funding:** We would like to acknowledge funding support from the Possehl Foundation (D.R. A.L.S. and P.F.; C180212; https://www.possehl-stiftung.de/de/index.html), John Fell OUP Research Fund (A.L.S.; 133/061; https://researchsupport.admin.ox.ac.uk/), the European Research Council (G.L.; ERC-2013-StG 337574-UNDEAD; https://erc.europa.eu/) and the Natural Environment Research Council (G.L.; NE/H005269/1 & NE/K005243/1; https://nerc.ukri.org/). During parts of this work A.L.S. was also funded by Biotechnology and Biological Sciences Research Council (BB/K004468/1 and BB/K001388/1; https://bbsrc.ukri.org/). The funders had no role in study design, data collection and analysis, decision to publish, or preparation of the manuscript.

**Competing interests:** The authors have declared that no competing interests exist.

contexts. Since the prevalence of medieval STH infections mirror those in modern endemic countries the factors affecting STH decline in Europe may also inform modern intervention campaigns.

## Author summary

Parasitic helminths (worms) are important infections of humans in many less well developed countries, particularly those in tropical and sub-tropical regions. These infections are not a major problem in modern Europe but parasite eggs are readily detected in archaeological contexts. To estimate a key epidemiological parameter, the prevalence of infection, we examined large numbers of single graves from Medieval Europe and found that the rates of infection with two soil transmitted nematodes (*Ascaris* spp. and *Trichuris trichiura*) were as prevalent as in many modern endemic areas. We also identified two cestodes that humans acquire from eating undercooked red meat (*Taenia* spp.) or freshwater fish (*Diphyllobothrium latum*). Using prevalence and ancient DNA data we explored helminth epidemiology in Medieval European populations and factors that may influence infection including age, sex, sanitation, hygiene and culinary practices. The Medieval prevalence rates provide a historical baseline for Europe and an interesting comparator for modern epidemiological studies in other parts of the world. It is noteworthy that helminths were endemic in historical Europe but were eradicated prior to the development of modern drugs. In this sense studying changes in helminth prevalence in historical Europe may provide insights into control efforts in modern endemic regions.

## Introduction

Intestinal helminths have afflicted humans throughout history and their robust eggs are detectable in a wide range of archaeological contexts [1–7]. In modern times soil transmitted helminths (STH) affect more than 1.5 billion people worldwide, mostly in Less Economically Developed Countries [8–10]. Estimations of the burden of STH infections vary considerably with some of the best regional and global estimates provided by meta-analyses assessing prevalence [8–10]. Population-wide prevalence rates vary according to environment, socio-economic factors, as well as cultural practices and the degree of urbanisation [11, 12]. Although rarely life threatening, STH infections are responsible for substantial morbidity in affected populations, being clinically associated with anaemia, malnutrition and developmental effects. Children consistently suffer from greater morbidity, with higher prevalence and intensity of infection than older age groups [13, 14]. Adult infections represent a reservoir for post treatment reintroduction of STH to children and the potential impact of community wide treatments has received considerable attention [15, 16].

Effective anthelmintic drugs were first introduced in the early 1960s and represent the primary intervention for modern control efforts although this is often supported by provision of clean water, improved sanitation and health education programmes [17–20]. Most areas with endemic helminth infections have experienced some attempt at intervention and the World Health Organization (WHO) recommends that 75% of school age children in at risk regions should regularly receive chemotherapy. Despite these efforts helminths remain endemic in many parts of the world. Perhaps the most notable modern eradication successes were achieved in Japan and South Korea through a combination of intensive anthelminthic

campaigns and extensive infrastructure improvement programs [21–23]. In modern Europe STH infections are very rare yet there are many reports of infection in archaeological deposits (reviewed in [24, 25]).Within Europe the decline in STH infections pre-dates the development of modern anthelminthic drugs and studying historic European populations provides an interesting comparator to studies in modern endemic areas.

Helminth eggs are robust and can be identified in a wide range of archaeological samples including the abdominal region of mummified bodies and skeletal remains as well as communal waste pits or latrines using microscopic techniques (e.g. [2, 6, 26, 27]). Egg morphology is adequate for genus-level diagnosis, while molecular approaches can be employed to identify species [1, 7, 28, 29]. The two nematodes *Ascaris* sp. and *Trichuris trichiura* are both among the most prevalent STH in modern populations and also widely reported in archaeological sites (e.g. [7, 30]). A wide range of other helminths that infect humans have been detected in archaeological samples including *Schistosoma* spp., *Enterobius vermicularis*, *Taenia* spp. and *Diphyllobothrium latum* [24, 31]. The detection of *Diphyllobothrium* and *Taenia* eggs in archaeological samples has been used to comment on the dietary and culinary practices of past populations since transmission to humans involves consumption of uncooked or undercooked fish or red meat [7]. Unfortunately, most reports of parasites in archaeological deposits deal with few individuals or communal deposits, therefore the prevalence of these infections in historic Europe remains unclear.

The large single-grave dataset allowed estimation of the prevalence of helminth infections in Medieval Europe, representing a period prior to the development of modern control methods. To determine the rates of helminth infection in Medieval Europe we analysed 589 single graves from seven medieval graveyards located in three European countries (UK, Germany and Czech Republic). Metadata associated with these sites allowed us to assess the influence of potential risk factors including age, sex and the estimated population size of the community. The term prevalence of infection usually refers to a living population, representing the proportion of infected individuals at a specific time. Since the samples under study were obtained from the pelvic (sacral) region of skeletal remains we report prevalence as the number/percentage of burials positive for helminth eggs. In some sites these rates could be sub-divided according to time, skeletal age or sex. To contextualise our results, we compared the data obtained from Medieval Europe with those from modern endemic regions [8–10]. These data provide a pre-modern-intervention baseline for comparison with other analyses of helminth epidemiology in historical and modern contexts.

## Materials and methods

### Sample material

589 pelvic soil samples were analysed in this study. A comprehensive anthropological analysis was available for the majority of samples. The sites included in this study are the north-eastern suburb of the early medieval stronghold of Břeclav-Pohansko (Czech Republic, 97 samples, [32]), Ellwangen-Jagst (Germany, 204 samples), Rottenburg Sülchenkirche (Germany, 91 samples, [33]), Ipswich Stoke Quay (UK, 83 samples), All Saints in the Marsh, Peasholme; the former Haymarket, York (UK, 35 samples), Worcester Cathedral (UK, 65 samples) and Brno Vídeňská street (Czech Republic, 14 samples). The excavating archaeologists collected the samples on site from the pelvic area of buried bodies. Samples from communal Medieval deposits in Bristol (4 samples) and Lübeck (9 samples) were part of a previous study [7] and included as a comparator for molecular analyses. Although "control" (e.g. sediment from the skull) were not analysed for all graves this was performed for a subset of samples. We also analysed sediment taken from the subsurface and a similar depth to the skeletal level from some

graveyards, and non-graveyard associated sediment. In all cases these "control" sediments contained no helminth eggs or parasite aDNA. The remaining unprocessed material is archived at the Department of Zoology, University of Oxford.

## aDNA extraction

Soil subsamples (~5 g) were re-hydrated in 20 ml volumes with overnight incubation at room temperature with gentle agitation (Titer-Tek plate shaker, setting 3/10; Titertek-Berthold, Pforzheim, Germany). Of this ~500μl was retained for microscopic analysis (see below). To enrich for parasite eggs for the aDNA extraction the samples were sieved through a series of disposable nylon sieves with decreasing aperture size (1030 μm, 500 μm, 100 μm; Plastok Associates Ltd, Birkenhead, UK), centrifuged (400 g 10 min) and re-suspended in 1 ml of TE buffer (Qiagen, Hilden, Germany). These were homogenised with 1mm glass beads in a BeadBeater (BSP BioSpec, Bartlesville, USA) and aDNA extracted using Qiagen Blood&Tissue kit (Qiagen, Hilden, Germany).

## Microscopic diagnosis

Aliquots of the rehydrated subsample were analysed without enrichment for microscopy as this may have led to differential recovery. Microscopy was undertaken using a Nikon Eclipse E400 with Nikon 10x/0.25 Ph1 DL and 40x/0.65 Ph2 DL lenses (Nikon UK Ltd., Kingston-Upon-Thames, UK). Photographs were recorded on a QImaging MP5.0 RTV camera (QImaging, Surrey BC, Canada) using the 40x lens (Nikon). Parasite egg counts were extrapolated from replicated counts to the initial dry weight. The subsample used for diagnosis was discarded after microscopy and not used for aDNA isolation.

## Sample handling and preparation workflow

Ancient DNA handling practices have been outlined in a range of publications [34–36]. A specialised workflow was developed to prevent any contamination of samples or aDNA by aDNA extracts, PCR amplicons or unprocessed samples (as described in [7]). None of the parasites targeted in this study are endemic in the UK or any country where material was received from. None of the laboratories where samples have been processed have ever handled or stored modern samples containing these parasites, hence modern contamination is unlikely.

The workflow was strictly unidirectional for both material and researchers with no equipment or consumables transferred back between steps or the three physically separate locations where work was undertaken. The laboratory where archaeological samples were initially handled and where aDNA was extracted was an extraction clean laboratory (no PCR, no large quantities of DNA handled). The processing was further confined to a dedicated still air hood (UV2 PCR, Ultra-Violet Products Ltd, Cambridge, UK). The workspace in the still air hood was treated to remove contaminants using a combination of UV irradiation (2 x 30min) and ChemGene HLD4L (Medimark Scientific Ltd, Sevenoaks, UK). Decontamination was performed prior to and after any handling of samples. All PCR was performed in a physically separate, dedicated laboratory space. Where PCR re-amplification was performed these were set up in another dedicated hood (in a second laboratory) decontaminated as described above. All agarose gel analyses and further processing of products occurred in a dedicated space within a third laboratory.

## PCR, aDNA sequencing and genetic diversity

The PCR strategy and the primers used targeting *T. trichiura* ITS-1 are described in [7]. Sequencing was run at the Wellcome Trust Centre for Human Genetics and the Department

of Zoology at the University of Oxford. Sequences were aligned using MEGA7 [37]. MiSeq pair read data was assembled and de-replicated using USEARCH v8 [38].

Samples that yielded at least 100 sequence reads were subsampled to various sequence depths (10 repeats each, subsampled to 50, 100, 150, 200, 250, 500, 1000, 1500, 2000, 2500, 3000, 4000, 5000, 6000, 7000, 8000, 9000, and 10000 reads, for grave samples the maximum sub-sampling level was 6000 as no samples yielded 7000 reads) using USEARCH v8 [38]. To calculate species richness, the R package iNEXT (iNterpolation and EXTrapolation, [39]) was used within R 3.4.2, [40] and RStudio [41].

## Results

### Sites and contextual data

To determine the rates of helminth infection in Medieval Europe we analysed 589 samples associated with the abdominal region of skeletons from 7 graveyards dating between 680 CE and 1700 CE (Fig 1). Age and sex were estimated according to established methods based on the skeletal remains [42]. The medieval data was compared with modern data extracted from three meta-analyses assessing global prevalence of helminths between 1973 and 2010 [8–10].

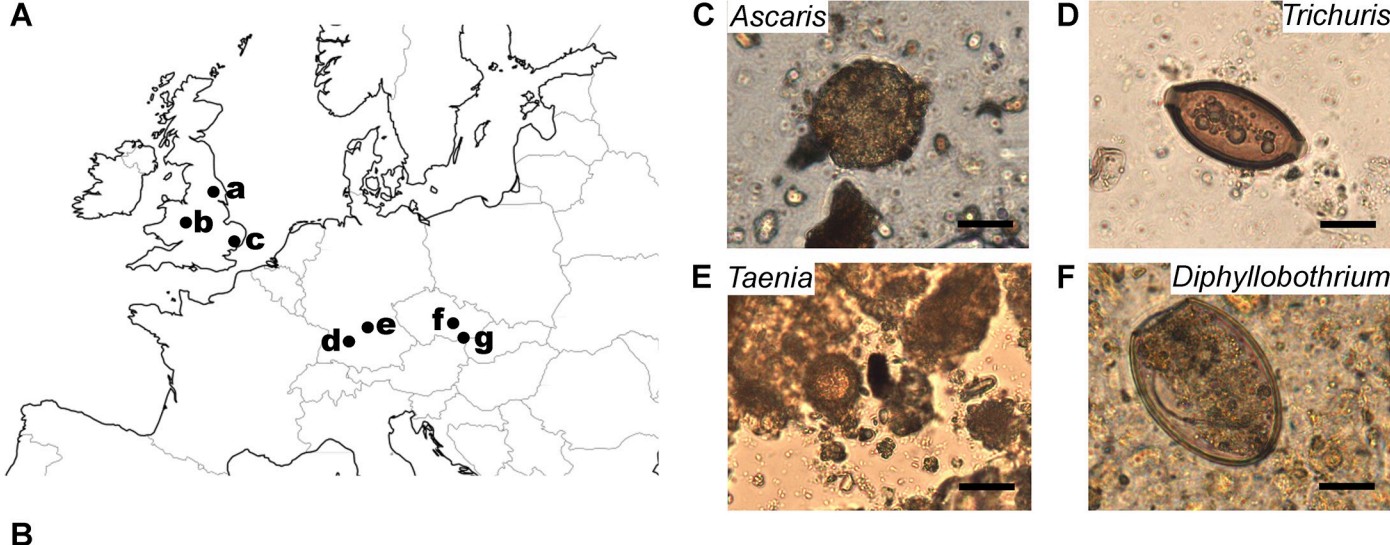

| | Site | Country | Samples (n) | Dating | Ascaris [%] | Trichuris [%] | Taenia [%] | Diphyllobothrium [%] |
|---|---|---|---|---|---|---|---|---|
| a | York | UK | 35 | 1100-1600 | 37.1 | 8.6 | 0 | 0 |
| b | Worcester | UK | 65 | 680-1100 | 9.2 | 1.5 | 0 | 0 |
| c | Ipswich | UK | 83 | 900-1500 | 28.9 | 6.0 | 7.2 | 6.0 |
| d | Rottenburg | DE | 91 | 700-1500 | 22.0 | 4.4 | 9.9 | 2.2 |
| e | Ellwangen | DE | 204 | 700-1700 | 22.1 | 7.4 | 2.5 | 1.0 |
| f | Pohansko | CZ | 97 | 875-950 | 35.1 | 18.6 | 2.1 | 2.1 |
| g | Brno | CZ | 14 | c1400 | 42.9 | 28.6 | 0 | 0 |

**Fig 1. The sites, samples and parasites.** A map showing the location of sampled sites (A); Summary table providing background information on each site and overall percentages of each parasite (B). Representative photomicrographs of the eggs detected in this study (C: *Ascaris*, D: *Trichuris*, E: *Taenia* and F: *Diphyllobothrium*, scale bar: 20 μm). The map represented in A was modified from the NASA SEDAC centre https://sedac.ciesin.columbia.edu/maps/gallery/search?facets=region: europe&facets=theme:water.

## Prevalence of helminth infections in Medieval Europe

Helminth eggs were detected by microscopic analysis of samples from the pelvic (sacral) region of skeletons. The eggs exhibited well-preserved diagnostic features and four helminths (*Trichuris*, *Ascaris*, *Taenia* and *Diphyllobothrium*) were identified (representative micrographs in Fig 1C–1F). Eggs from the faecal-oral transmitted nematodes *Trichuris* and *Ascaris* were the most common and detected in all sites, although the prevalence varied between sites. In all seven sites more individuals were infected with *Ascaris* than *Trichuris*. The site with lowest rates of *Ascaris* and/or *Trichuris* infection was Worcester (UK, 9.2% and 1.5%, respectively) with all other sites having substantially higher rates of infection (Fig 1B). Combining the data from all sites, *Ascaris* eggs were identified in 25.1% of individuals (range 9.2–42.9%, σ = 10.5%) and *Trichuris* eggs identified in 8.5% of individuals (range 1.5–28.6%, σ = 8.8%). In contrast, eggs from the food-associated cestodes *Taenia* and *Diphyllobothrium* were detected in samples from four sites (Ellwangen, Pohansko, Rottenburg and Ipswich). Within sites where the food-derived cestodes were present, *Taenia* spp. eggs were detected in 3.5% of individuals (range 2.1–9.9%) with *Diphyllobothrium* eggs detected in 1.1% of individuals (range 1.0–6.0%).

Co-infection with two or more parasites was detected in 38 of the 589 samples (6.4%). The most frequent co-infection was *Trichuris* and *Ascaris* (3.9% of samples; odds ratio 2.35 95% CI 1.29–4.26), followed by *Ascaris* and *Taenia* (1.2% of samples; odds ratio 1.51 95% CI 0.6–3.83). Infection with more than two parasites was only seen in three samples (one each of *Ascaris-Trichuris-Taenia*, *Ascaris-Trichuris-Diphyllobothrium* and *Ascaris-Trichuris-Taenia-Diphyllobothrium*).

## The prevalence rates of *Ascaris* and *Trichuris* in Medieval Europe were comparable to those in modern endemic regions

Although human infection with *Ascaris* and *Trichuris* are almost completely absent (and mostly travel related) in modern European populations, these infections are endemic in many developing countries. The prevalence rates within Medieval European sites were comparable to those reported for regions of the world where soil transmitted helminths are considered endemic (Fig 2). The modern prevalence rates were calculated from data presented in three meta-analyses covering the periods 1973 to 1993 [8], 1994–2003 [9] and 2004–2010 [10]. For both *Ascaris* and *Trichuris* the mean global prevalence of infection in endemic countries has reduced in recent years with fewer regions reporting prevalence rates of greater than 20% in the 2004–2010 reporting period compared with earlier periods. The medieval European prevalence rates for *Ascaris* infection (mean 25.1% of individuals, range 9.2–42.9%, σ = 10.5%) were most similar to those reported by Chan et al. [8] and de Silva et al. [9], but significantly higher (p = 0.00014, ANOVA) than those reported by Pullan et al. [10] (Fig 2A). In contrast, the prevalence rates for *Trichuris* infection in our medieval samples (mean 8.5% of individuals, range 1.5–28.6%, σ = 8.8%) were not significantly different to the rates reported by Pullan et al. [10] or De Silva et al. [9], but significantly lower (p = 0.00027, ANOVA) than those reported in Chan et al. [8] (Fig 2B).

## Prevalence rates in males and females

Five of the medieval sites provided identification of sex for the majority of adult skeletons (Fig 3). The analysis of sex-based prevalence rates was restricted to adults since osteological sex identification is considered unreliable for sub-adults [42]. Osteological data on skeletal sex was available for 378 individuals, with a slightly larger proportion of males (57.1%, n = 216) than

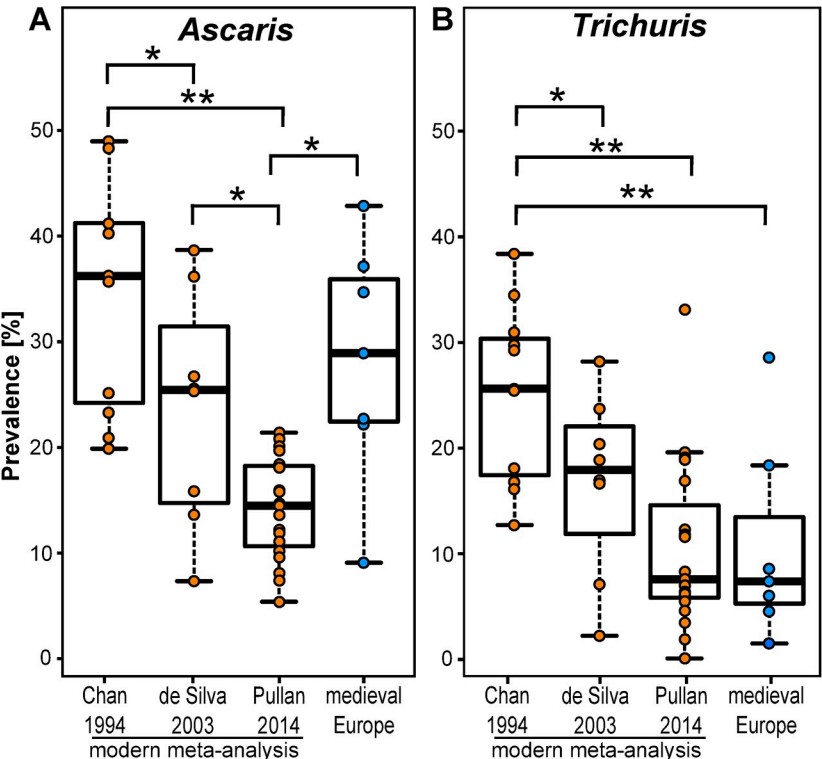

**Fig 2. Comparison of prevalence rates of *Ascaris* and *Trichuris* between modern and medieval populations.**
Prevalence rates of *Ascaris* (A) and *Trichuris* (B) identified in archaeological sites and comparison with regional
prevalence rates published in meta-analyses by Chan 1994 [8], de Silva 2003 [9] and Pullan 2014 [10]. Orange dots
indicate prevalence rates in different endemic regions. Blue dots indicate prevalence rates in each of the archaeological
sites within Europe. Significant differences between groups indicated by * p<0.05, ** p<0.005.

females (42.9%, n = 162) identified. The overall prevalence of parasitic infection was not significantly different between the two sexes (35.8% in females and 29.1% in males; Fisher's Exact two-tailed p-value 0.1825). There was also no evidence for male-to-female bias with the prevalence any of the individual parasites (Fisher's Exact test, female-to-male ratio, 6.8%:7.9% for *Trichuris*, p = 0.843; 25.9%:20.8% for *Ascaris*, p = 0.267; 4.9%:4.2% for *Taenia*, p = 0.804 and 4.3%:1.9% for *Diphyllobothrium*, p = 0.217; Fig 3). There was a degree of variability between sites (Fig 3A), but this was often observed in cases where overall numbers of infected individuals were low within a particular site (Fig 3B).

## Prevalence rates by age at death

With modern datasets the prevalence of infection is often reported as differing between preschool age (<5 years), school age (5–18 years) and adults, although prevalence has a weaker association with age than intensity of infection [43]. The single grave datasets allowed for an estimate of prevalence according to age at death in five sites (Pohansko, Ellwangen, Rottenburg, Ipswich and Worcester), and the skeletal remains from York were subdivided into juvenile and adult. The detailed age structure of the combined sample set and at each site is given in Supplementary S1 Fig. Age-prevalence data was categorised as infant/young child (<5 years), child (6–18 years), adult (18–40 years) and older adult (40+ years). Combining data from all sites within a mixed effects linear model revealed a significantly lower prevalence of infection with *Ascaris* in the youngest age category (<5 years) compared with the 6–18 years

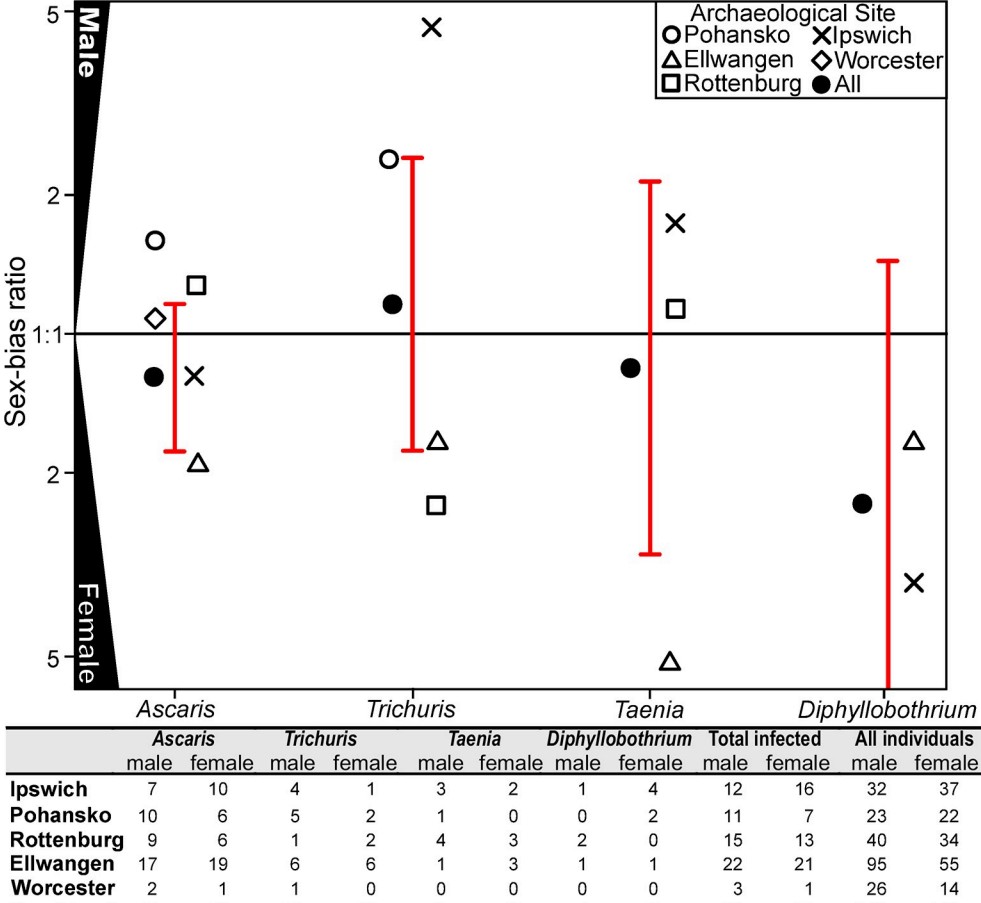

| | Ascaris | | Trichuris | | Taenia | | Diphyllobothrium | | Total infected | | All individuals | |
|---|---|---|---|---|---|---|---|---|---|---|---|---|
| | male | female | male | female | male | female | male | female | male | female | male | female |
| **Ipswich** | 7 | 10 | 4 | 1 | 3 | 2 | 1 | 4 | 12 | 16 | 32 | 37 |
| **Pohansko** | 10 | 6 | 5 | 2 | 1 | 0 | 0 | 2 | 11 | 7 | 23 | 22 |
| **Rottenburg** | 9 | 6 | 1 | 2 | 4 | 3 | 2 | 0 | 15 | 13 | 40 | 34 |
| **Ellwangen** | 17 | 19 | 6 | 6 | 1 | 3 | 1 | 1 | 22 | 21 | 95 | 55 |
| **Worcester** | 2 | 1 | 1 | 0 | 0 | 0 | 0 | 0 | 3 | 1 | 26 | 14 |
| **Combined** | 45 | 42 | 17 | 11 | 9 | 8 | 4 | 7 | 63 | 58 | 216 | 162 |

**Fig 3. The prevalence of helminth infection in males and females.** The ratio of infected male to female individuals for each of the four parasites, *Ascaris*, *Trichuris*, *Taenia* and *Diphyllobothrium*. Each of the sites indicated by a different symbol. The actual numbers of males and females infected with each parasite are given in the Table below the graph as is the cumulative total of all sites and the total numbers of males and females identified at each site. The bars represent the 95% confidence interval of all sites and no significant differences were identified according to sex.

old (p = 0.004) or 40+ years (p = 0.009) groups and a significantly higher prevalence of infection with *Ascaris* in the 6–18 age group compared to the 18–40 years old (p = 0.026, Fig 4). In contrast, there was no evidence for age associated changes in the prevalence of infection with *Trichuris* (Fig 4B). For both nematode infections there was considerable variation between sites with some exhibiting greater differences between age groups and other sites exhibiting smaller, or no, differences between age groups. It is noteworthy that the prevalence rates for *Ascaris* were higher within the child group compared with all other age groups in three of the five sites (Pohansko, Ipswich and Worcester). The 35 skeletal remains from York were categorised with much lower resolution as juvenile (n = 6) and adult (n = 29). Despite the bias in skeletal age groups a far higher proportion of juveniles (100%) were positive for *Ascaris* infection than adults (24.1%). The prevalence of *Trichuris* was also higher in juveniles (16.1%) than in adults (6.9%).

Four sites contained cestode eggs (*Taenia* and/or *Diphyllobothrium*) and although the prevalence rates were much lower than those for nematodes (Fig 1F) no cestode eggs were detected in the 1–5 age group (n = 54) compared with 4 positives in the 6–18 group (5.9%; n = 68), 15 in the 18–40 age group (6.6%; n = 227) and 14 in the over 40 year olds (11.9%; n = 118).

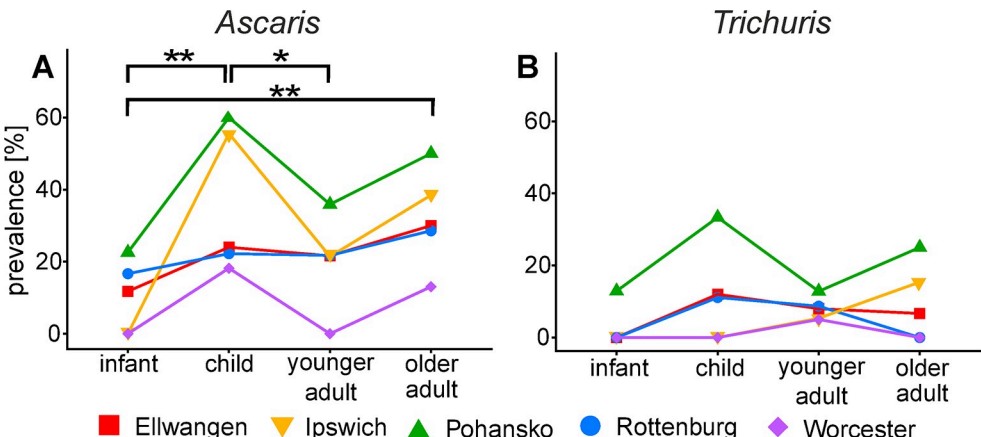

**Fig 4. The prevalence of nematode infections in different age groups.** The prevalence of *Ascaris* (A) and *Trichuris* (B) in different age groups at five medieval sites. Infant 0–5 years, Child 6–18 years, Younger Adult 19–40 years, Older adult 40+years. Each site is depicted with different symbols and colours according to the key. Significant differences in the overall rates of infection are indicated by horizontal bars * p<0.05, ** p<0.005.

### Prevalence rates by population size or historical era

The prevalence rates for infection with STH (*Ascaris* and *Trichuris*) may have been influenced by differences in the size of the local population. To test this, we plotted the total prevalence rates for *Ascaris* or *Trichuris* infection against estimated population sizes of the sampled localities (Fig 5A and 5B). The highest rates for prevalence of infection with *Ascaris* were in York (UK) and Brno (CZ), which also had the highest population estimates for the sampled time. This pattern was not evident for the *Trichuris* prevalence rates. To avoid the potential confounding effect of differences in age structure of the sampled population from each site the relationship between *Ascaris* infection and population size was re-examined focussing on adults (Fig 5C and 5D). Although Brno remained the site with highest prevalence the adult prevalence rates in York were more similar to those seen with samples from Ipswich, Rottenburg and Ellwangen which had much lower population size estimates. Hence, no correlation could be identified between prevalence rates and the size of the local population.

Two sites contained large numbers of samples over extended time periods; Ellwangen, with 203 dated samples between the 7th and 18th centuries and Ipswich, with 78 dated samples between the 9th and 15th centuries. Within Ellwangen (Fig 6) and Ipswich (Supplementary S2 Fig) the prevalence rates for nematode infections (*Ascaris* and *Trichuris*) remained stable over time. The prevalence rates for cestode infections were much lower than with nematodes. Although there were no significant changes in cestode prevalence over time it is noteworthy that in Ellwangen, *Diphyllobothrium* was only detected in samples dating prior to the 13th century (n = 106 compared with n = 96). Also, in Ellwangen, *Taenia* eggs were not detected in 81 samples dating between the 11th and 14th centuries whereas they were detected between the 7th and 10th centuries (n = 65) and the 15th to 18th centuries (n = 56).

### Ancient DNA sequencing and genetic diversity

Ancient DNA (aDNA) was extracted from 182 samples which contained parasite eggs (samples from Ipswich, Pohansko, Brno, Ellwangen and Rottenburg). The 182 aDNA extracts were subject to PCR and MiSeq sequencing for detection of *Trichuris trichiura* ITS-1 (*Tt*ITS-1) and *Ascaris* spp. COX-1 fragments using the protocols described in Flammer et al. [7]. Forty-nine single grave derived samples (3 from Ipswich, 18 from Pohansko, 2 from Brno, 17 from

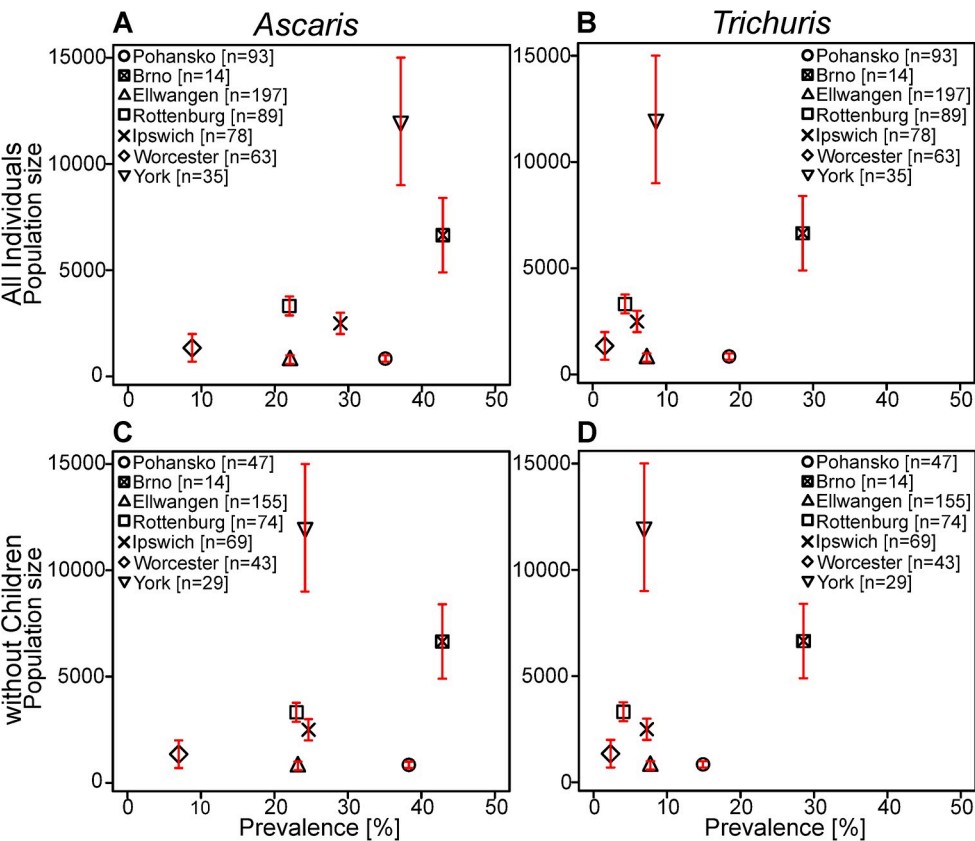

**Fig 5. The influence of population size on parasite prevalence.** The population size of the different locations was estimated from historical records and plotted against the prevalence of *Ascaris* (A and C) or *Trichuris* (B and D). Panels A and C represent the whole data set. Panels B and D represent data from adult groups (>18 years or identified as adult). Each site is identified with a different symbol according to the insert. The numbers of samples considered with each site are given in the insert. The error bars represent the range of estimated population size for each group.

Ellwangen and 9 from Rottenburg) yielded at least one positive parasite sequence for *T. trichiura* ITS-1 and/or *Ascaris* COX-1. Six samples generated more than 100 non-singleton reads covering the hypervariable *Tt*ITS-1 fragment which were used to estimate the diversity of sequences in single grave-derived samples (Fig 7; Po 1–5 and Ip 1). The level of diversity in single grave derived samples was compared with the diversity of sequences obtained from 13 samples from communal deposits from medieval Bristol (UK) and Lübeck (DE) [7]. Each of the six single grave samples were dominated by a single sequence although the dominant sequence differed between samples. In three samples the dominant sequence was identical (Po 1, 2 and 5) and also the dominant sequence in all four communal samples from medieval Bristol (Br). This sequence was detected in 7 of 8 latrine samples from Lübeck (Lu) but was only dominant in 3 of those samples (Lu 2, 3 and 8). The dominant sequence in Po4 was identified in Bristol (Br1 and 2) and the dominant sequence in Po3 was present in Br2. One of the rarer *Tt*ITS-1 sequences in Po1 (yellow) was the dominant sequence in one of the Lubeck communal samples (Lu9). The sharing of sequences between sites or samples means little other than some haplotypes were more widely distributed than others. However, the fact that different individuals harboured *T. trichiura* with different ITS-1 sequences did allow us to consider the level of diversity in single skeleton-associated samples compared with communal samples.

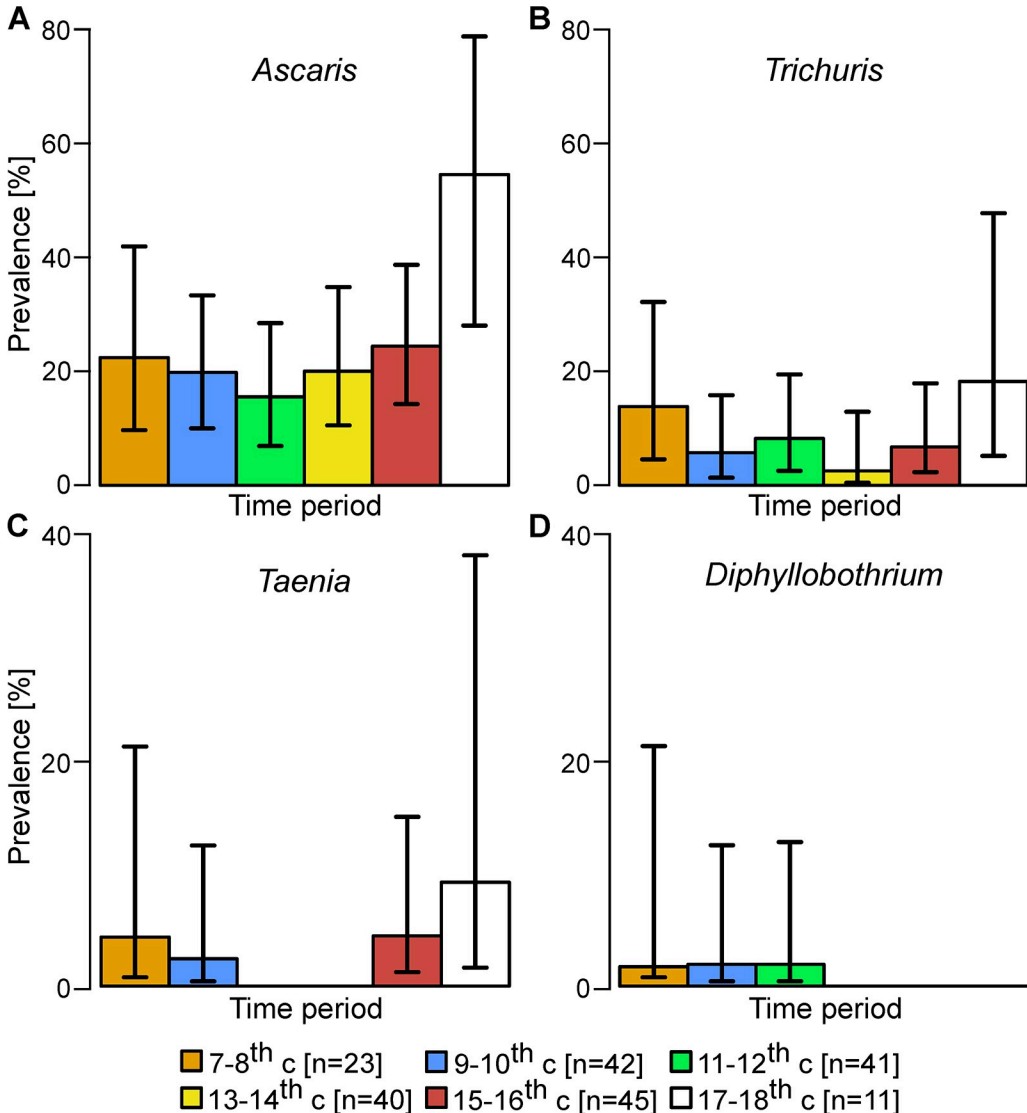

**Fig 6. The prevalence of helminth infections in Ellwangen over time.** The prevalence rates for *Ascaris* (A), *Trichuris* (B), *Taenia* (C) and *Diphyllobothrium* (D) infection in Ellwangen were segregated according to 6 time periods from the 7th-8thc to the 17th-18thc as indicated. The number of samples in each time period is also identified. Bars represent the proportion of infected individuals and error bars represent 95% confidence intervals.

When considering diversity within and between sample types we only included samples with at least 100 non-singleton reads (6 single grave and 13 communal samples from our previous study). The mean and median number of unique *Tt*ITS-1 fragment sequences (represented at greater than 1% of total reads) for single graves was considerably lower (mean 4.0/median 3.0) than with communal samples (mean 8.3/median 9.0). To assess genetic diversity by species richness, the sequencing depth was corrected for by down-sampling to various levels (10 repeats each, subsampled to 50, 100, 150, 200, 250, 500, 1000, 1500, 2000, 2500, 3000, 4000, 5000, 6000, 7000, 8000, 9000, and 10000 reads, for grave samples the maximum sub-sampling level was 6000 as no samples yielded 7000 reads). Species richness for the single grave samples was not only lower than in communal deposits, but also did not increase with increasing read depth (Fig 7B and 7C). The diversity of *Tt*ITS-1 fragment sequences in a sample can be

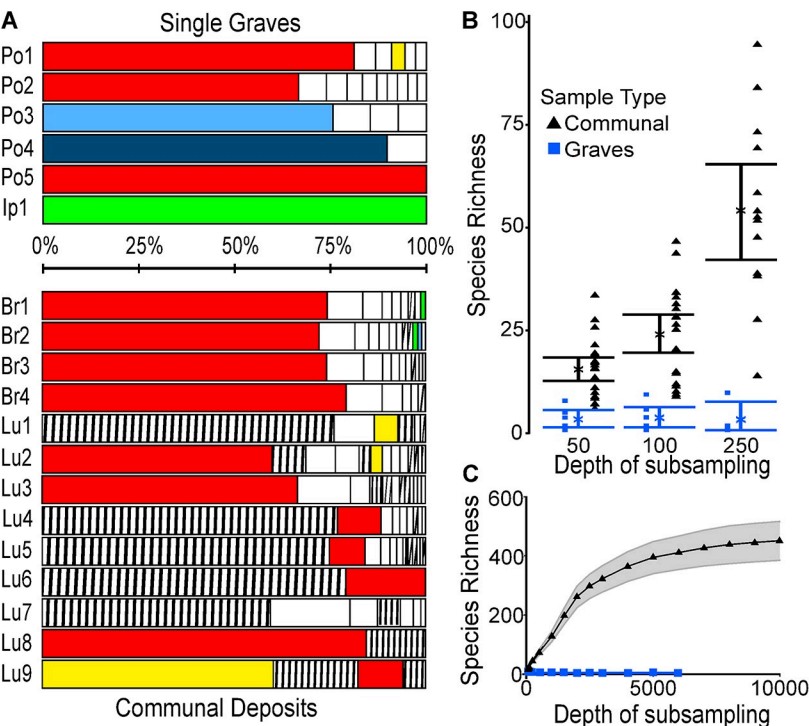

**Fig 7. The diversity of *Trichuris trichiura* ITS1 fragment sequences within single grave samples and contemporary communal deposits.** A stacked bar chart (A) indicating the proportion of different sequences in each of 6 single grave samples (5 from Pohansko, Po and one from Ipswich, Ip) and 13 communal deposits (4 from Bristol, Br1-4; and 9 from Lübeck, Lu1-9) where greater than 100 sequences/sample were obtained. The colours represent sequences that were identified in multiple samples with solid colours representing the widespread group 1 sequences (identified in reference 7) and the striped bars group 2 sequences that were common in Lübeck, rare in Bristol and not found elsewhere (7). Segments with no colour represent sequences identified in a single sample. The species richness of samples (B and C) segregated by sample type (single grave, blue) and communal deposit (black). The diversity of all samples are represented in panel B with the mean species richness ± Standard Error of the mean depicted at different depths of sub-sampling between 50 and 250 sequences. Higher depths of subsampling are presented in panel C to reveal the plateau of the sampled diversity, the mean diversity (line) with 95% confidence intervals (shaded region) are depicted.

estimated from the plateau level obtained across different read depths, for single graves this was approximately 4 whereas for communal samples the plateau was estimated to be approximately 450. These differences in diversity estimates exceeded the difference in numbers of parasite eggs seen in single grave (negative samples not included, mean 125.6±92.2 eggs/g) than communal deposits (mean 1135±986.6 eggs/g).

## Discussion

Helminth parasites have afflicted humans throughout history and remain a significant burden in many parts of the world [43, 44]. Extensive anthelminthic treatment campaigns have been active for decades in many parts of the world, with the aim of reducing the global burden of soil transmitted helminths (STH, [43, 45, 46]). STH infections are very rare in modern Europe however, eggs from two of the key STH nematodes, *Ascaris* spp. and *Trichuris trichiura* are often identified in archaeological deposits [7, 47–50]. Most reports of parasite eggs in archaeological deposits either represent a small number individual skeletons or communal deposits such as latrines or waste pits [1, 3–5, 51, 52]. Understanding the epidemiology of infection in past populations requires the analysis of large individual-based datasets [53] and burial-associated sediments represent a potential source of such information.

The prevalence of infection is a key epidemiological measure that could be important in understanding the dynamics of infection in Medieval Europe. Prevalence of parasite eggs was estimated by analysis of samples from the pelvic (sacral) region of 589 skeletal remains from seven locations in the UK, Germany and the Czech Republic, dated between 680 CE and 1700 CE. This large dataset is important in both historical and modern contexts, providing key information on the pervasiveness of intestinal parasitism in Medieval Europe before the introduction of modern anthelminthic treatments. The impact of infections on populations is related to the prevalence and intensity of infection rather than their presence or absence in a location, hence large individual-based datasets are much more informative than occasional samples.

Four helminths were detected in this study, two faecal-orally transmitted nematodes (*Ascaris* spp. and *Trichuris trichiura*) and two food-transmitted cestodes (*Diphyllobothrium latum* and *Taenia* spp.). Each of these allows us to comment upon various aspects of life in the medieval period in Europe including levels of sanitation and culinary or dietary habits.

The prevalence rates of *Ascaris* spp. and *Trichuris trichiura* in medieval Europe were comparable to those reported in areas where these infections remain endemic. For *Ascaris* spp. the prevalence in medieval Europe was most similar to the higher levels reported in 1994 ([8] and Fig 2A). For *T. trichiura* prevalence in medieval Europe was most similar to the more recent (lower) levels of infection ([10] and Fig 2B). Interestingly, the rate of embryonation for *T. trichiura* is more sensitive to lower temperatures than for *Ascaris* [54] which might explain the lower prevalence of *T. trichiura* compared with *Ascaris* in medieval Europe. However, in at least some communal deposits, including house-associated latrines in Medieval Lübeck (DE), *T. trichiura* was more common than *Ascaris* spp. [7]. It is therefore reasonable to conclude that there were no environmental limitations to transmission of either of these nematodes in medieval Europe, with similar factors affecting the rate of transmission of these nematodes (e.g. sanitation, hygiene, behaviour) in modern populations without modern anthelmintic drugs. The rates of co-infection with these two nematodes in medieval Europe were comparable to those reported in modern endemic sites [55].

The prevalence rates we report should be considered an underestimate of the prevalence in the living population due to the need to positively detect the low numbers of eggs that remain associated with the abdominal region of skeletal remains. A combination of local disturbance and degradation may further reduce the sensitivity of detection although both *Ascaris* and *Trichuris* eggs are readily detected in much older samples (e.g. [5, 7, 51, 56, 57]). The relatively high prevalence of nematode STH infections in medieval Europe may relate to increased urbanisation without adequate hygiene or practices such as the use of night soil to fertilise crops (discussed in [24]).

The change from STH being common infections in Medieval Europe to the very low, non-endemic levels seen in modern Europe raises the question of when and how this change occurred. The medieval prevalence rates suggest that Europe was not a special case where the parasites were at the extremes of their range, at low prevalence rates and easier to eradicate. The drive towards non-endemic infection with STH in modern Europe is likely to be the result of human intervention, such as improved sanitation and hygiene. Whilst sanitation in the 20th century was considerably improved from the medieval period, parasite eggs were detected in two of three skeletons [3] and a latrine associated with trench sites from the 1914–18 conflict [58]. Whether these examples were reflective of the broader population is debatable and may represent imported parasites or high transmission rates due to the unhygienic conditions associated with trench warfare. It is likely that the reduction in STH infections within Europe would have been highly dependent on local conditions and more work is required to establish the patterns of infection in time and space as well as factors contributing to the decline of

enteric helminths. In his seminal paper, "This Wormy World" Norman Stoll [59] indicated that STH and cestode infections were present in Europe during the interwar period although, as Stoll acknowledged, these estimates rely on extensive extrapolation and, even if accurate, may reflect poorer regions with less infrastructural development.

More data is required to link reductions to specific events or interventions but this could be a fruitful area for future consideration. Indeed, in modern STH endemic countries the effects of anthelminthic deworming are best sustained in conjunction with an improved programme of water, sanitation and hygiene (WASH, [18, 60, 61]). For example, the helminth control successes in Japan, the Republic of Korea and Taiwan, came about through government provision of hygiene infrastructure, health education, alongside wide scale chemotherapy [21, 23, 62–65]. In other parts of the world where infrastructural improvements were less dramatic the prevalence rates post-treatment communities often rebounds to pre-treatment levels [66] which probably relates to the effect of untreated adults and the environmental resilience of eggs [46, 67]. Notably, the reductions in infections in Europe were achieved without modern chemotherapeutic drugs, illustrating the power of infrastructural and other societal improvements.

The eggs of two cestodes (*Taenia* spp. and *D. latum*) were identified in four sites although the rates of infection were much lower than with the nematodes. Eggs from these cestodes have previously been identified in archaeological deposits [5, 7, 30, 57, 58, 68–71] although before the current study estimates of prevalence would have been inappropriate. Human infections with *Taenia* spp. and *D. latum* are derived from raw or undercooked food, specifically red meat (pork or beef) and freshwater fish, respectively. Of note, in the current study, in all sites where cestode eggs were found (Ipswich, Rottenburg, Ellwangen and Pohansko), both *Taenia* spp. and *D. latum* were detected, but only one individual harboured a co-infection. Moreover, Ipswich and Rottenburg contained the highest prevalence levels for both cestodes. The pattern of infection suggests that the availability of certain food types or particular culinary practices (i.e. under-cooking of red meat and/or freshwater fish) may have been more common in some locations.

For five sites metadata indicating the sex and a predicted age at death allowed us to consider these as potential risk factors for infection with helminths. There was no overall male/female bias in infection rates for any of the four parasites detected in this study although in some sites there were differences in infection rates (e.g. Ipswich for *T. trichiura*). Both the lack of overall effect of sex and variation between sites is comparable to modern data sets [72]. Where sex bias has been reported in modern data sets including, the high levels of *T. trichiura* in young males from indigenous Shuar of Amazonian Ecuador [73] and Vietnamese female agricultural workers [74], these are thought to be driven by differential exposure due to cultural factors.

Age is commonly associated with changes in prevalence and intensity of parasite infections. With STH the most heavily infected group tends to be school-age children, which also experience the greatest pathological impacts (malnutrition, stunting, anaemia and intellectual retardation) [75]. The increased childhood infection risk has been attributed behavioural exposure to infection or lack of acquired immunity [46, 76]. In our medieval datasets, the prevalence of infection with *Ascaris* was highest in the 6–18 year old age category, but there was no overall age structure with *T. trichiura* infections. This is similar to modern datasets where the age association of STH prevalence is stronger with *Ascaris* than *Trichuris* (e.g. [77]). Modern anti-STH campaigns focus on the treatment of the medically most impacted school aged cohorts, although the role of adults in re-infection of post-treatment children is hotly debated (e.g. [45, 46, 60, 78]). Our data indicates that adults may have been an important source of infection in medieval societies; indeed they may have been instrumental for the exchange of different parasite strains between locations. The food-derived cestodes *Taenia* spp. and *D. latum* exhibited a

different pattern of infection, with the number of infected individuals increasing with age. This pattern might be expected if the probability of infection increased according to the amount of undercooked meat/fish consumed.

We considered whether there were any changes in the prevalence of any of the helminth infections over time within sites and whether the estimated population size at each of the medieval locations might affect parasite prevalence. Neither of these factors were associated with clear changes in the pattern of infection.

To assess diversity of *T. trichiura* in archaeological deposits we have previously developed a targeted approach using a diverse fragment of *Tt*ITS-1 [7]. Despite the very low numbers of eggs in single grave deposits we identified *T. trichiura* sequences in 36 samples with seven containing over 100 reads. All of the sequences obtained confirmed the presence of *T. trichiura* rather than other related *Trichuris* spp. Although, no communal deposit samples were available from the locations where single grave samples were available we were able to compare the diversity of *Tt*ITS-1 fragment with medieval samples from Lübeck (DE) and Bristol (UK) [7]. Each of the seven single grave samples with over 100 reads contained one highly dominant sequence, and in three samples this was the sequence we had identified in our previous study as the "core" group 1 sequence [7]. Interestingly with other grave samples the dominant (most frequent) sequences were also present, albeit at low levels in communal samples. As expected, the diversity of *Tt*ITS-1 sequences obtained from single graves was much lower than the diversity we detected in samples from communal deposits (from Lübeck and Bristol), as more people would have contributed to such a deposit. The maximum species richness values obtained for *Tt*ITS-1 diversity in single graves was 4 compared with over 450 in communal samples. The 100-fold difference cannot be explained solely by a lower number of eggs in single graves, as this difference is much lower and may be a measure of the number of adult worms contributing to a sample (by virtue of multiple people contributing to the communal latrine deposits). Unfortunately, there is no experimental or field data available comparing detectable *Tt*ITS-1 fragment diversity with egg numbers, worm load or sample type in modern circumstances, thus any further interpretation would be unwise.

In conclusion, we have determined the prevalence of four helminth parasites in the abdominal region of 589 medieval skeletons and compared these with modern prevalence rates. The presence and prevalence of two food derived cestodes (*Taenia* spp. and *D. latum*) provides valuable information on diet and culinary practices in the past. The prevalence of STH (*T. trichiura* and *Ascaris* spp.) in medieval Europe were comparable to the rates in modern endemic tropical and sub-tropical locations. Interestingly, these rates support the premise that these parasites circulated in Europe with a similar epidemiology to that seen in modern endemic countries and that in European history these infections were effectively reduced to non-endemic levels prior to the development of modern anthelminthic drugs. The reduced endemicity was most likely associated with changes in water, hygiene, agricultural practices and sanitation systems. Indeed, improvements to water supplies, sanitation and hygiene are often discussed with reference to modern control efforts [79, 80]. The European STH experience supports the idea that these WASH measures can be successful in the absence of modern anthelminthic drugs even with high rates of infection in the pre-intervention communities.

Even in the modern era it remains unclear whether chemotherapy alone is sufficient to control STH and it is therefore appropriate to explore other measures including WASH more thoroughly [16, 18, 60, 61]. This study represents the first large-scale analysis of the prevalence of helminths in medieval Europe and will act as a reference point for future archaeological studies across time and space. We also propose that the use of historical helminth datasets can be useful in contextualising modern epidemiology, particularly with respect to interventions.

## Supporting information

**S1 Fig. Age-associated prevalence of *Ascaris* and *Trichuris*.** Age-associated prevalence for the common nematode helminths *Trichuris* (A) and *Ascaris* (B) was analysed within and across sites. The age structure (C) varied considerably across the sampled sites.
(TIF)

**S2 Fig. The prevalence of helminth infections in Ipswich over time.** The prevalence rates for *Ascaris*, *Trichuris*, *Taenia*, and *Diphyllobothrium* infection in Ipswich were segregated according to 3 time periods: 9th-10th c, 11th-12th c and 12th-15thc. The number of samples in each time period is also identified. Bars represent the proportion of infected individuals and error bars represent 95% confidence intervals.
(TIF)

## Author Contributions

**Conceptualization:** Patrik G. Flammer, Mark Pollard, Greger Larson, Adrian L. Smith.

**Data curation:** Patrik G. Flammer, Hannah Ryan.

**Formal analysis:** Patrik G. Flammer, Hannah Ryan, Stephen G. Preston, Adrian L. Smith.

**Funding acquisition:** Dirk Rieger, Greger Larson, Adrian L. Smith.

**Investigation:** Patrik G. Flammer, Hannah Ryan, Sylvia Warren, Renáta Přichystalová, Rainer Weiss, Valerie Palmowski, Sonja Boschert, Katarina Fellgiebel, Isabelle Jasch-Boley, Madita-Sophie Kairies, Ernst Rümmele, Dirk Rieger, Beate Schmid, Ben Reeves, Rebecca Nicholson, Louise Loe, Christopher Guy, Tony Waldron, Jiří Macháček, Joachim Wahl.

**Methodology:** Patrik G. Flammer, Hannah Ryan, Mark Pollard, Greger Larson, Adrian L. Smith.

**Project administration:** Greger Larson, Adrian L. Smith.

**Resources:** Renáta Přichystalová, Rainer Weiss, Ernst Rümmele, Dirk Rieger, Beate Schmid, Ben Reeves, Rebecca Nicholson, Louise Loe, Christopher Guy, Tony Waldron, Jiří Macháček, Joachim Wahl, Adrian L. Smith.

**Software:** Patrik G. Flammer, Stephen G. Preston.

**Supervision:** Mark Pollard, Greger Larson, Adrian L. Smith.

**Visualization:** Patrik G. Flammer, Hannah Ryan, Adrian L. Smith.

**Writing – original draft:** Patrik G. Flammer, Hannah Ryan, Greger Larson, Adrian L. Smith.

**Writing – review & editing:** Patrik G. Flammer, Hannah Ryan, Stephen G. Preston, Sylvia Warren, Renáta Přichystalová, Rainer Weiss, Valerie Palmowski, Sonja Boschert, Katarina Fellgiebel, Isabelle Jasch-Boley, Madita-Sophie Kairies, Ernst Rümmele, Dirk Rieger, Beate Schmid, Ben Reeves, Rebecca Nicholson, Louise Loe, Christopher Guy, Tony Waldron, Jiří Macháček, Joachim Wahl, Mark Pollard, Greger Larson, Adrian L. Smith.

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
