## [Decision Letter · Decision Letter 0]

4 Jun 2020

Dear Prof Smith,

Thank you very much for submitting your manuscript "Epidemiological insights from a large-scale investigation of intestinal helminths in Medieval Europe." for consideration at PLOS Neglected Tropical Diseases. As with all papers reviewed by the journal, your manuscript was reviewed by members of the editorial board and by several independent reviewers. The reviewers appreciated the attention to an important topic. Based on the reviews, we are likely to accept this manuscript for publication, providing that you modify the manuscript according to the review recommendations. 

Two reviewers of your manuscript recommended to accept it after minor revision and I concur. Please see the reviewers comments' in detail and address them point-by-point basis. I expect your revised manuscript in time.

Sincerely,

jong-Yil Chai

Guest Editor

Mar Siles-Lucas

Deputy Editor

Two reviewers of your manuscript recommended to accept it after minor revision and I concur. Please see the reviewers comments' in detail and address them point-by-point basis. I expect your revised manuscript in time.

Reviewer's Responses to Questions

**Key Review Criteria Required for Acceptance?**

**Methods**

-Are the objectives of the study clearly articulated with a clear testable hypothesis stated?

-Is the study design appropriate to address the stated objectives?

-Is the population clearly described and appropriate for the hypothesis being tested?

-Is the sample size sufficient to ensure adequate power to address the hypothesis being tested?

-Were correct statistical analysis used to support conclusions?

-Are there concerns about ethical or regulatory requirements being met?

Reviewer #1: If the authors were to further define their methods, then this paper would be an especially valuable source. How were the eggs concentrated for analysis? Were the slides examined without concentrating the eggs? Were the sediment samples collected from the sacra? Were control samples analyzed from the cemeteries? Why were the microscopic preparations discarded? Where are the initial samples now stored?

Reviewer #2: The methods are clearly defined, however there are located after the discussion section, it is necessary to move them after the results section. The study design is appropriate as well as the sample size.

**Results**

-Does the analysis presented match the analysis plan?

-Are the results clearly and completely presented?

-Are the figures (Tables, Images) of sufficient quality for clarity?

Reviewer #1: The results are well-organized and succinct

Reviewer #2: Results are clear and following the study design. The figures are sufficient but a couple of tables could help the results presentation and summary, for example the prevalence of helminths in the different study sites.

**Conclusions**

-Are the conclusions supported by the data presented?

-Are the limitations of analysis clearly described?

-Do the authors discuss how these data can be helpful to advance our understanding of the topic under study?

-Is public health relevance addressed?

Reviewer #1: The conclusions and discussion are adequate. 

I think that basic scholarship could be improved by citing the following:

Camacho et a1 (2018) Recovering parasites from mummies and coprolites: an epidemiological approach. Parasites & Vectors (2018) 11:248

https://doi.org/10.1186/s13071-018-2729-4

This paper addresses and introduces a paleoepidemiological approach to archaeological parasitology.

Leles et al (2010) A parasitological paradox: Why is ascarid infection so rare in the prehistoric Americas? J Archaeol Sci. 2010;37:1510–20.

This paper summarized all reports from Europe to assess the commonness of geohelminth infection compared to other prehistoric records and developed reasons for the high soil transmitted parasite experience in Europe.

Replace Reinhard KJ. Parasitology as an Interpretive Tool in Archaeology. Am Antiquity. 644 1992;57(2):231-45 

with Reinhard, K., 2017. Reestablishing rigor in archaeological parasitology. International journal of paleopathology, 19, pp.124-134.

The 1992 reference is antiquated and the 2017 paper takes on issues that are current.

Reviewer #2: The conclusions are supported by the data and analysis undertaken. Limitations are mentioned in some parts but not clearly expressed. In some parts, such as page 24 line 387, there is a repetition of results which is not needed.

**Editorial and Data Presentation Modifications?**

Reviewer #1: Here are optional suggestions for the manuscript.

“Prevalence”, in parasitology parlance, refers to the total number of individuals in a population are infected at a specific period of time, usually expressed as a percentage of the population. In the case of this paper, prevalence is the number/percentage of interments positive for infection evidence. I suggest that prevalence as used by the authors be defined in the Introduction.

Reviewer #2: Minor English revision is needed in the document

**Summary and General Comments**

Reviewer #1: This paper could be published as is. It is a very significant, novel approach to assessing paleoepidemiology for Europe. It represents a monumental amount of work and innovative integration of standard microscopy and molecular biology. It established cemetery analysis as a statistically reliable manner of collecting hundreds of data points. Thus, based on this research, burial analysis takes its place alongside shaft feature analysis as key approach to defining infection across cultures, time and geography. It fits into an emerging realization that, for the Eurasian continent, Europe had a unique parasite pattern defined by geohelminths. Geographically, this pattern is not found in central Russia and westward. Therefore, Europe was an endemic area characterized by remarkably lax sanitation and hygiene controls.

Reviewer #2: In summary is well designed, implemented and presented. However, there are some statements that need further evidence or references for support, such as the one in lines 549-464. The study does provide with helpful epidemiological insights to intestinal helminths that could be useful in certain endemic regions.

PLOS authors have the option to publish the peer review history of their article (what does this mean?). If published, this will include your full peer review and any attached files.

Reviewer #1: Yes: Karl J. Reinhard

Reviewer #2: No
---

## [Decision Letter · Decision Letter 1]

14 Jul 2020

Dear Prof Smith,

We are pleased to inform you that your manuscript 'Epidemiological insights from a large-scale investigation of intestinal helminths in Medieval Europe.' has been provisionally accepted for publication in PLOS Neglected Tropical Diseases.

Best regards,

jong-Yil Chai

Guest Editor

Mar Siles-Lucas

Deputy Editor

Your revised manuscript has been reviewed by referees. Both of them recommended to accept your manuscript as it is and I concur. Your cooperation with PLoS NTD is highly appreciated.

Reviewer's Responses to Questions

**Key Review Criteria Required for Acceptance?**

**Methods**

-Are the objectives of the study clearly articulated with a clear testable hypothesis stated?

-Is the study design appropriate to address the stated objectives?

-Is the population clearly described and appropriate for the hypothesis being tested?

-Is the sample size sufficient to ensure adequate power to address the hypothesis being tested?

-Were correct statistical analysis used to support conclusions?

-Are there concerns about ethical or regulatory requirements being met?

Reviewer #1: The revised paper includes improvements of the methods description.

Reviewer #2: The observations from the methods section were correctly addressed

**Results**

-Does the analysis presented match the analysis plan?

-Are the results clearly and completely presented?

-Are the figures (Tables, Images) of sufficient quality for clarity?

Reviewer #1: The results are clear and concise.

Reviewer #2: The observations from the results section were correctly answered

**Conclusions**

-Are the conclusions supported by the data presented?

-Are the limitations of analysis clearly described?

-Do the authors discuss how these data can be helpful to advance our understanding of the topic under study?

-Is public health relevance addressed?

Reviewer #1: The conclusions address all four of these goals successfully.

Reviewer #2: The observations made in this section were correctly addressed

**Editorial and Data Presentation Modifications?**

Reviewer #1: The revision needs no modification od data presentation.

Reviewer #2: Accept

**Summary and General Comments**

Reviewer #1: The revision addresses the reviewer comments perfectly.

Reviewer #2: There are no new comments

PLOS authors have the option to publish the peer review history of their article (what does this mean?). If published, this will include your full peer review and any attached files.

Reviewer #1: **Yes: **Karl J Reinhard

Reviewer #2: No

---

## [Editor Report · Acceptance letter]

3 Aug 2020

Dear Prof Smith,

We are delighted to inform you that your manuscript, "Epidemiological insights from a large-scale investigation of intestinal helminths in Medieval Europe.," has been formally accepted for publication in PLOS Neglected Tropical Diseases.

Best regards,

Shaden Kamhawi

co-Editor-in-Chief

Paul Brindley

co-Editor-in-Chief
